# Impact on All-Cause and Cardiovascular Mortality Rates of Coronary Artery Calcifications Detected during Organized, Low-Dose, Computed-Tomography Screening for Lung Cancer: Systematic Literature Review and Meta-Analysis

**DOI:** 10.3390/cancers13071553

**Published:** 2021-03-28

**Authors:** Sébastien Gendarme, Helene Goussault, Jean-Baptiste Assié, Cherifa Taleb, Christos Chouaïd, Thierry Landre

**Affiliations:** 1Service de Pneumologie, Centre Hospitalier Intercommunal de Créteil, 94010 Créteil, France; Helene.goussault@chicreteil.fr (H.G.); jean-baptiste.assie@inserm.fr (J.-B.A.); Christos.Chouaid@chicreteil.fr (C.C.); 2Inserm U955, UPEC, IMRB, équipe CEpiA, 94010 Créteil, France; 3Functional Genomics of Solid Tumours Laboratory, Centre de Recherche des Cordeliers, Inserm, Sorbonne Université, Université de Paris, 75013 Paris, France; 4Service de gériatrie, Hôpital René Muret, APHP, 93270 Sevran, France; cherifa.taleb@aphp.fr; 5UCOG, Hôpital René Muret, APHP, 93270 Sevran, France; thierry.landre@aphp.fr

**Keywords:** cardiovascular mortality, coronary artery calcification, lung cancer screening, meta-analysis

## Abstract

**Simple Summary:**

The results of several randomized studies showed the efficacy of organized, low-dose, computed-tomography (CT) scan lung-cancer screening in lowering all-cause and lung-cancer-specific mortality rates. Low-dose CT scans can also detect and quantify coronary artery calcifications (CACs). By means of meta-analysis, we were able to show that the presence of CACs in CT performed in this setting was associated with an enhanced risk of cardiovascular and all-cause mortality for men and women. These finding plead for the implementation of preventive interactions against cardiovascular risk in lung-cancer screening-program participants found to have CACs.

**Abstract:**

Although organized, low-dose, computed-tomography (CT) scan lung-cancer screening has been shown to lower all-cause and lung-cancer-specific mortality, the primary cause of death for subjects eligible for such screening remains cardiovascular (CV) mortality. This meta-analysis study was undertaken to evaluate the impact of screening-scan-detected coronary artery calcifications (CACs) on CV and all-cause mortality. We conducted a systematic review and meta-analysis of studies reporting CV mortality according to the Agatson CAC score for participants in a lung-cancer screening program of randomized clinical or cohort studies. PubMed, Embase, and Cochrane databases were screened in June 2020. Two authors independently selected articles and extracted data. Six studies, including 20,175 subjects, were retained. CV and all-cause mortality rates were higher for subjects with CAC scores >0, with respective relative risks of 2.02 [95% CI 1.23–3.32] and 2.29 [95% CI 1.00–5.21]. Both mortality rates were even higher for those with high CAC scores (>400 or >1000). CACs are more common in men than in women, with an odds ratio of 1.49 [95% CI 1.40–1.59]. The presence of CAC is associated with CV mortality with an RR of 2.05 [95% CI 1.20–3.57] in men and 2.37 [CI 95% 1.29–5.09] in women, respectively. Analysis of lung-cancer-screening scans for CACs is a tool able to predict CV mortality. Prospective studies within those programs are needed to assess the benefit of primary CV prevention based on CAC detection.

## 1. Introduction

Results of the American NLST and the European NELSON studies showed the efficacy of organized, low-dose, computed-tomography (CT) scan screening for lung cancer of smokers or ex-smokers [1,2]. All-cause and specific lung-cancer mortality rates, respectively, were 6.8% and 20% lower in the NLST study [1], and specific mortality was 24% lower in the NELSON study [2]. According to the latter, the benefit could be even greater for women, with 39–61% lower lung-cancer-specific mortality. Those findings led the Unites States and numerous other countries to initiate organized lung-cancer-screening programs [3,4].

Those studies also showed the notable impact on all-cause mortality of other pathologies associated with tobacco, namely, cardiovascular (CV) events. Indeed, in the NLST study, more than 50% of deaths were attributed to CV events [1]. Smoking is also a major risk factor in CV mortality, thereby explaining the relationship between those pathologies and lung cancer [5,6]. All those findings highlighted the interest of combining organized, low-dose, CT scan lung-cancer screening with a more general search for tobacco-associated diseases [7], especially targeting chronic obstructive pulmonary disease (COPD) and CV events, aptly called “Big-3” screening [8].

The benefit of organized screening for CV prevention in this population of smokers and ex-smokers has rarely been studied and not yet demonstrated. In theory, CV prevention seems clearly amenable to organized screening. Identified risk factors are modifiable at the individual level, and a decrease in their exposure leads to decrease in risk [9]. Moreover, the existence of an asymptomatic or latent phase, during which interventions can effectively change the natural evolution of the disease [10]. Indeed, no test—with an acceptable cost—has demonstrated sufficient ability to justify CV diseases screening. The performance of thoracic low-dose CT scans to identify CACs, as a tool able to predict CV events, was validated in several studies [11,12], despite the diminished spatial resolution of low-dose scan slices [13,14,15]. Although low-dose CT scans require a low radiation dose (<1.5 mSv) and is not ECG-gated, they can detect CAC with good agreement (r = 0.89) in comparison with conventional ECG-gated CT protocol and depending on the attenuation and thickness of the slices [16]. The Agatson score enables quantification of lesions containing calcium deposits and risk stratification [17].

Therefore, we conducted a systematic literature review and a meta-analysis to determine the efficacy of measuring CACs, visualized on low-dose CT scans, as a tool to predict CV and all-cause mortality in subjects participating in an organized program screening for lung cancer.

## 2. Methods

### 2.1. Literature Search

The literature review consisted of searching PubMed, Embase, and Cochrane databases using the following terms: “(lung cancer OR lung carcinoma OR pulmonary carcinoma [MeSH Major Topic]) AND (calcification OR arterial calcification OR aortic calcification [MeSH Major Topic])”. Two authors (S.G. and T.L.) ran the search on 1 June 2020. We also considered the references cited in the main reviews published on the subject, the gray literature identified by searching congress abstracts and scientific communications, and by contacting experts in the field.

Articles, published in English or French, were selected according to 2015 PRISMA-P (Preferred Reporting Items for Systematic Reviews and Meta-Analyses) recommendations. After exclusion of doubles, the same two authors independently examined the titles and abstracts. The inclusion criteria for selected papers were: Subjects eligible for lung-cancer screening, i.e., smokers or ex-smokers, not diagnosed with lung cancer and without signs suggestive of lung cancer at inclusion, who underwent thoracic, low-dose, CT scan in the framework of a randomized or non-randomized clinical trial, or cohort studies on organized screening for lung cancer. CV and/or all-cause mortality rates were given in each report as a function of the presence or absence of CACs in the population or expressed as relative risk (RR), hazard ratio (HR), or odds ratio (OR). Cross-sectional studies, case reports, or case series were excluded from the analysis (Appendix A), as well as ECG-gated CT protocol and extra-coronary calcification detection studies. Then, the two authors independently read the full texts of each article and extracted the data of interest: Sample size, participants characteristics (age, sex, body mass index), duration of smoking, number of current and former smokers (>2 or 4 weeks since smoking cessation, according to studies), hypertension status, diabetes status, Agatston CAC score, and CV events at inclusion and during follow-up (myocardial infarction, angina, CV death, congestive heart failure, revascularization of coronary arteries). Divergences were resolved by consensus or the decision of a third author (C.C.).

The methodological quality of the selected studies was evaluated (S.G. and T.L.) using the grid published by Downs and Black [18] that enabled assessment of randomized and non-randomized trials. This evaluation concerned the quality of the study, its internal and external validities, management of selection and measurement biases and confounding factors.

### 2.2. Statistical Analyses

The primary endpoint was CV and all-cause mortality rates as according presence or absence of CAC. Secondary endpoints were the mortality rates according to a high Agatson score (>400 or >1000, depending on the study) and sex.

CV mortality is defined by diseases of the circulatory system (according to International Classification of Diseases-10-CM Diagnostic Code I00-99), which mainly includes hypertensive diseases, ischemic heart diseases, pulmonary circulation diseases, and cerebrovascular and artery diseases.

For studies reporting CV or all-cause mortality risk stratified according to the different Agatson scores, a pooled risk was obtained for all CAC classes > 0. The results are expressed as RR [95% confidence interval (CI)] for a random-effect (for *I*^2^ > 50%) or fixed-effect model (for *I*^2^ < 50%) as a function of heterogeneity. The weight of each study was estimated by the inverse variance of the fixed-effect model and with the Paule–Mandel estimator for the random-effect model [19]. Heterogeneity of the studies was analyzed visually and with the *I*^2^ heterogeneity test, with *I*^2^ > 50% supporting significant heterogeneity of the studies included in the meta-analysis. Publication bias was evaluated with a funnel plot for each meta-analysis. All statistical analyses were computed with R software version 3.6.3. and its metagen library [20].

## 3. Results

Among the 1913 references identified in Medline and Cochrane databases and other sources (Figure 1), 1794 articles did not meet inclusion criteria on reading the title, and 91 articles on reading the abstract. Twenty-eight articles were selected for full-text reading. Twenty-two were excluded: Nine concerned populations that did not participate in an organized lung-cancer screening program; three each were doubles, did not report mortality data, or had not stratified the CAC quantification score; and two each examined extra coronary artery calcifications or had used ECG-guided CT scans.

Only six studies fulfilled the inclusion/exclusion criteria, reporting CV and/or global mortality rates for 20,175 subjects according to their CAC scores [21,22,23,24,25,26]; five were randomized clinical trials on organized screening for lung cancer, and the last was a cohort study (Table 1). These studies enrolled 57% men and 47% active smokers. The Agatson CAC score was determined in five of them [21,22,23,24,25] and the last applied a visual quantitative score [26] with excellent agreement with the Agatson score using non-parametric concordance test (R = 0.84, *p* < 0.001). The CAC frequency ranged from 46% to 76%, depending on the study, with a high score for 7–19% of the participants. No study was excluded after assessment of the methodological quality (Appendix A).

The globally symmetrical distribution of the funnel plots for each of the studies showed the absence of major publication bias (Appendix A). Among the 17,851 subjects for whom a CV mortality analysis was possible (Table 2A), CAC detection was associated with excess CV mortality, with RR estimated at 1.68 [95% CI 1.39–2.04] for the fixed-effect model, and 2.02 [95% CI 1.23–3.32] for the random-effect model that was retained in light of the significant heterogeneity among studies (*I*^2^: 73%, *p* = 0.01, Figure 2A). Among the 8621 participants for whom the all-cause mortality rate could be analyzed (Table 2B), CAC positivity was associated with a higher risk, with an RR of 1.47 [95% CI 1.34–1.62] for the fixed-effect model and 2.29 [95% CI 1.00–5.21] for the random-effect model (Figure 2B). Between-study heterogeneity was found with *I*^2^ at 82% and *p* < 0.01. Among the 16,445 subjects for whom CV mortality could be analyzed according to a high CAC score (>400 or >1000 or score > category 4) (Table 3A), CV mortality was higher, with RR at 2.55 [95% CI 1.70–3.84] for both the fixed effect model and the random-effect model (Figure 3A); notably, heterogeneity was absent: *I*^2^ = 0% et *p* = 0.68. Finally, for the 9780 subjects for whom all-cause mortality could be analyzed according to a high CAC score (Table 3B), all-cause mortality was also higher with RR at 2.47 [95% CI 1.75–3.47] for the fixed-effect model and 2.90 [95% CI 1.57–5.36] for the random-effect model (Figure 3B), with *I*^2^ at 35% indicating moderate heterogeneity among studies.

In studies reporting CAC scores according to sex [22,23,25,26], CAC frequencies and rate of high CAC score were significantly higher for men than women, respectively: 60% vs. 50% (*p* < 0.001) and 20% vs. 11% (*p* < 0.001) (Table 4). One study based on 5718 subjects reported CV mortality in both men and women according to CAC score [22]. In this study, the presence of CAC was associated with CV mortality with an RR of 2.05 [95% CI 1.20–3.57] in men and 2.37 [CI 95% 1.29–5.09] in women, respectively. Five studies described the number of CV events and CV mortality according to sex for 18,562 subjects (Table 5). The respective number of CV events and CV mortality for men had RRs of 5.55 [95% CI 2.65–11.61] and 2.11 [95% CI 1.01–4.39], with moderate global heterogeneity (*I*^2^ = 37% and *p* = 0.19) (Figure 4).

## 4. Discussion

This meta-analysis showed that CAC detection by thoracic low-dose CT scan during an organized lung-cancer screening program is able to predict CV and all-cause mortality rates in this population. This CV mortality risk was significantly higher for individuals with high CAC scores. CAC and elevated CAC score frequencies were significantly higher for men, but the predictive value of the CAC score on CV mortality did not seem to differ between men and women. These findings agree with reported observations—in other populations—of a significant link between CACs and CV and all-cause mortality rates [26,27,28,29].

Thus, in a prospective cohort of 9715 patients, including 8.3% with diabetes, the adjusted risk of death for the participants with most elevated CAC scores (>400) vs. those with no detected CACs (CAC = 0) increased by HRs of 4.64 (95% CI: 3.74–5.76) and 3.41 (95% CI: 2.22–5.22) for non-diabetics and diabetics, respectively [30]. Analysis of another cohort comprised of 4143 consecutive asymptomatic patients, at least 55 years old with no known coronary artery disease, followed for almost 15 years, showed that independently of smoking status, high CAC score was associated with a higher mortality. For any CAC, the non-smokers adjusted mortality risk was more than three-times higher (HR 3.07, 95% CI = 2.32–4.07, *p* < 0.001) and that for smokers it was almost five-time higher (HR 4.67, 95% CI = 3.52–6.20, *p* < 0.001) [27]. Patients without additional cardiac risk factors (such as hypertension, diabetes, dyslipidemia, family history of premature coronary artery disease) had similar findings: The adjusted risk of death increased incrementally according to CAC severity.

The prevalence of CACs reported in our meta-analysis (55%) is in agreement with previous reports; Ruparel et al. have reported a CAC prevalence of 62% in 680 subjects, current or ex-smokers aged 60–75 invited to a “lung health check” and undergoing ungated, non-contrast low-dose computed tomography [31]. At present, other studies are needed, especially cost–efficacy analyses, to establish which CAC score should serve as an indication for diagnostic or therapeutic intervention [8,18,32]. For CV-specific and all-cause mortality rates, comparable findings were reported for the absence of interaction with other causes of death, e.g., linked to chronic respiratory or neoplastic diseases [33]. CAC enable better prediction of CV events than some clinical scores [34,35]. However, even in the absence of CACs, almost 15% of individuals meeting the eligibility criteria for organized lung-cancer screening can experience a CV event [36].

The CAC predictive value for CV risk could differ between men and women. Indeed, in contrast to some reported observations [37], our meta-analysis revealed that men had a greater risk for CV events and CV mortality (RR: 2.86, 95% CI: 1.55–5.30) and a higher CAC frequency and more elevated Agaston scores than women. Other studies found CAC appeared in women more than 10 years later [22]. According to the NLST study, CAC frequency was higher for men (83%) than women (63%) [22]. The predictive value of the Agaston scores of 0–100, 100–1000, and >1000, respectively, were similar according to sex, with respective RRs of 1.51 (95% CI 0.88–2.59), 2.13 (95% CI 1.26–3.62), 2.76 (95% CI 1.56–4.88) for men and 1.87 (95% CI 1.04–3.36), 2.43 (95% CI 1.29–4.56), 3.74 (95% CI 1.62–8.63) for women. In this setting, the concomitant screening for lung cancer and CV risk assessment might optimize recommendations and management of stopping smoking and develop primary prevention and therapeutic intervention programs to prevent or slow the progression of tobacco-associated diseases [38,39].

This meta-analysis has several limitations. First, marked heterogeneity existed among the studies, mainly because of the results of the NELSON study [21]. The risks of CV-specific and all-cause mortality in that study were heightened even more, with an adjusted RR at 5.98 [95% CI 2.49–14.35]. The characteristics of their population, notably the higher percentage of men (83%), might explain that difference, at least in part. Second, classification of the Agatson CAC score varied depending on the study, with heterogeneity for the definition of the high threshold. Other types of scores, notably qualitative score or algorithms based on artificial intelligence, are being evaluated to establish a standardized analysis of the CAC score [40,41,42]. Compared to the Agaston score, an assessment taking into consideration the volume, density, and morphology of the CACs might better predict the CV risk [43].

Another limitation of this meta-analysis is the heterogeneity of participants’ characteristics inherent in the inclusion criteria for lung-cancer screening, depending on the study. Age, duration of smoking, the number of pack-years, and CV comorbidities differed between the populations included and could explain the heterogeneity of CAC frequencies and the predictive value for CV mortality. Furthermore, this meta-analysis was based on aggregated data, which did not allow subgroup analyses on individual subject’s information, notably examination of different CAC thresholds or different risk levels according to age, smoker status, or CV comorbidities.

Finally, it is not clear from the current data whether knowledge of the presence of CAC on a low-dose CT scan in this population could be translated in a decrease of CV mortality. Nevertheless, the ITALUNG study suggests that information about CAC presence in the screening report may be a factor reducing CV-specific mortality through further investigation and earlier management of CV diseases [24].

## 5. Conclusions

In these specific population of active and former smokers, CT-scan evaluation of CACs detected during organized lung-cancer screening was able to predict CV-specific and all-cause mortality risks and should support taking CV risk into account in these programs.

## Figures and Tables

**Figure 1 cancers-13-01553-f001:**
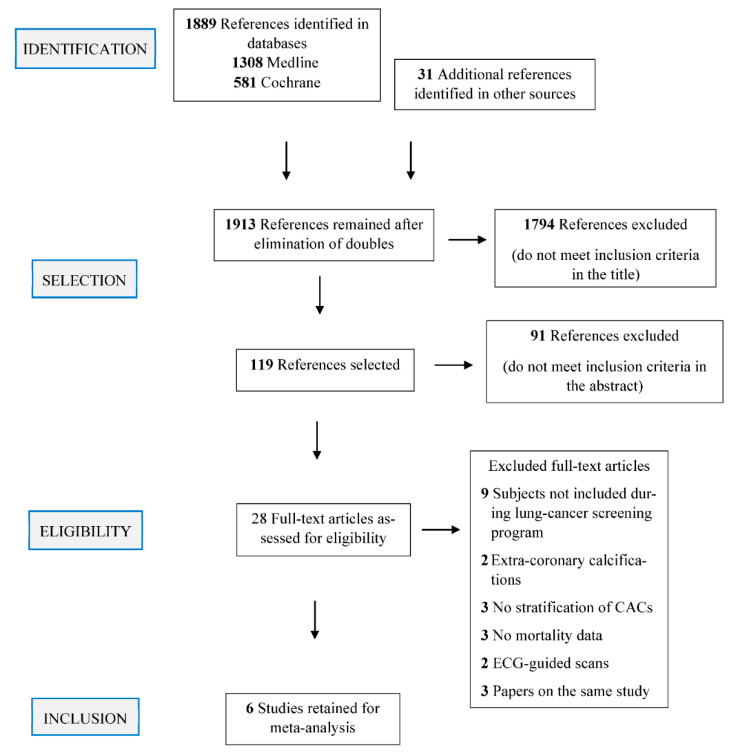
Flow chart for article selection.

**Figure 2 cancers-13-01553-f002:**
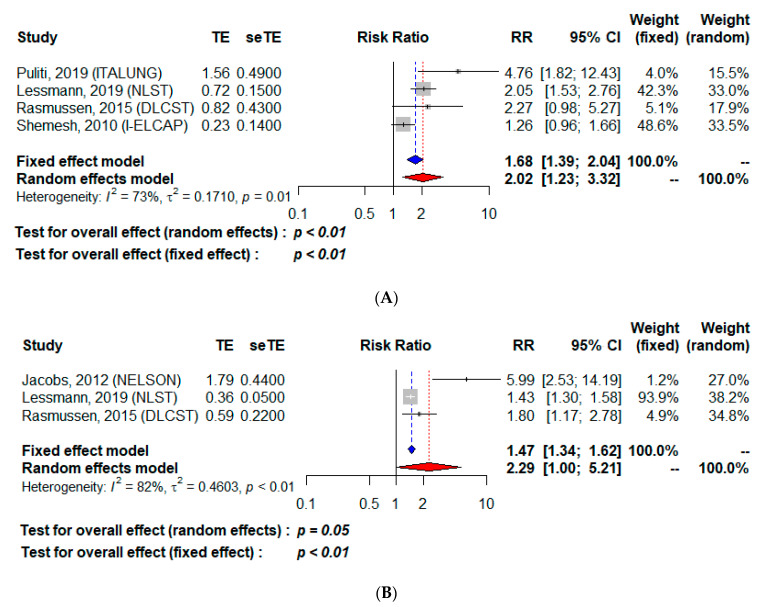
Cardiovascular (**A**) and all-cause (**B**) mortality according to the presence of coronary artery calcifications. TE, estimate of treatment effect, e.g., log hazard ratio or risk difference; seTE, Standard error of treatment estimate; RR, relative risk.

**Figure 3 cancers-13-01553-f003:**
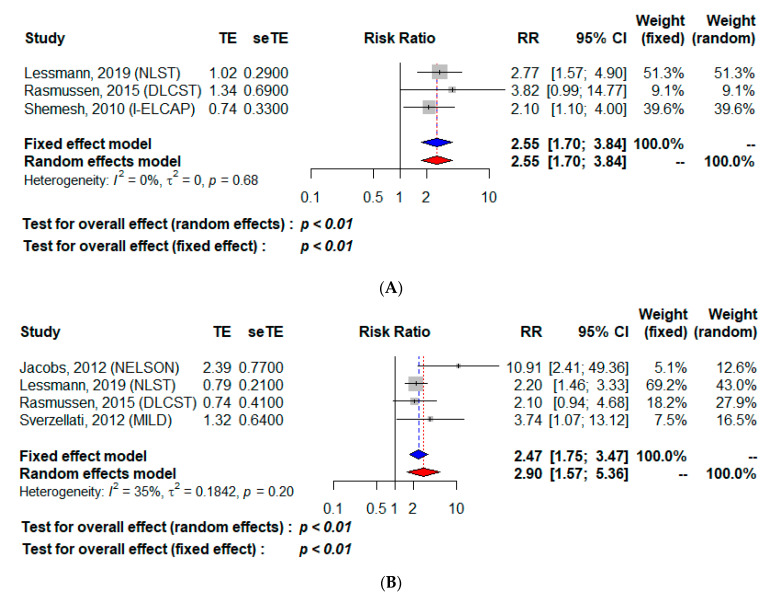
Cardiovascular (**A**) and all-cause (**B**) mortality according to a high coronary artery calcifications score. TE, estimate of treatment effect, e.g., log hazard ratio or risk difference; seTE, Standard error of treatment estimate; RR, relative risk.

**Figure 4 cancers-13-01553-f004:**
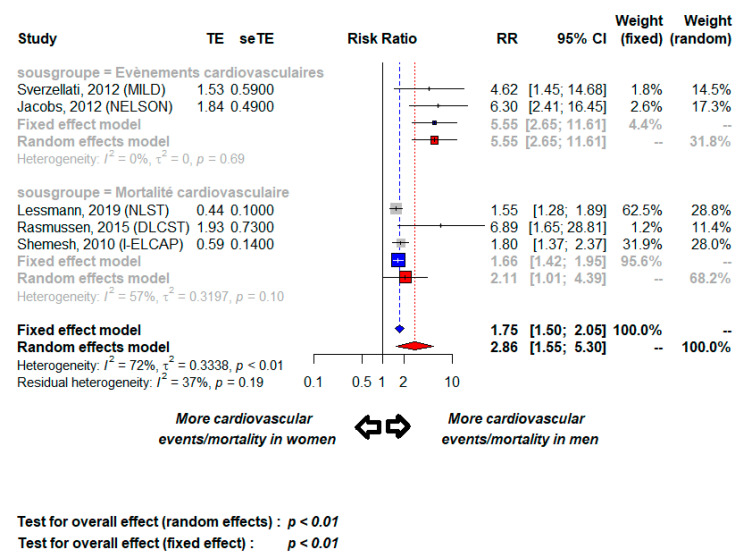
Cardiovascular events and mortality according to sex; TE, estimate of treatment effect, e.g., log hazard ratio or risk difference; seTE, Standard error of treatment estimate; RR, relative risk.

**Table 1 cancers-13-01553-t001:** Sociodemographic and clinical characteristics of the populations in studies retained for the meta-analysis.

Parameter	Jacobs [21]	Lessmann [22]	Sverzellati [23]	Shemesh [26]	Rasmussen [25]	Puliti [24]
Study name	NELSON	NLST	MILD	ELCAP	DLCST	ITALUNG
Countries	Belgium, Holland	United States	Italy	United States	Denmark	Italy
Type of study	Case–control niched in a clinical randomized trial	Case–control niched in a clinical randomized trial	Cohort niched in a clinical randomized trial	Cohort	Case–control niched in a clinical randomized trial	Cohort niched in a clinical randomized trial
Number of subjects	958	5718	1159	8782	1945	1613
Inclusion criteria						
Age range, years	50–74	55–74	49–75	40–85	50–70	55–69
Pack-years	>15 *	>30	>20	>10	>20	>20
Median follow-up, months	21.5	78	36	72.3	85.2	135.6
Clinical characteristics of the populations						
Men	83%	62.1%	68.4%	48.9%	55%	CAC +/−:73%/58%
BMI (kg/m^2^)	–	DS/C: 27.3/27.4	26.0	–	25	–
CV risk factor						
Active smoker, %	56%	DS/C: 58%/45%	65.1%	34%	76%	–
Mean pack/year	–	DS/C: 60/50	38.4	M/F: 0.7/45.6	34	CAC +/−: 42/39
Hypertension	64%	DS/C: 42%/34%	24.9%	–	14%	–
Diabetes	7%	DS/C: 18%/11%	6%	M/F: 9.2%/5.1%	2%	–
Hypercholesterolemia	75%	-	–	–	7%	-
No CACs	24%	24.5%	53.8%	40.7%	53%	54.3%
High CAC > score	17% ^a^	-	6.9% ^b^	18.7% ^b^	7% ^b^	–

* Or >15 cigarettes/day for > 25 years or > 10 cigarettes/day for >30 years, DS: Deceased subjects, C: Controls; Abbreviations BMI, body mass index; CV, cardiovascular; CAC +/−, coronary artery calcification positive or negative; ^a^ Agatson score >1000, ^b^ Agatson score > 400.

**Table 2 cancers-13-01553-t002:** Cardiovascular (CV) or global mortality according to coronary artery calcification (CAC) status.

Study	*N*	CV Mortality	CAC > 0		RR (CI 95%)		Adjusted Log RR (SD Log RR Adjusted)	Adjusted TO
			Events	Total	Non-adjusted	Adjusted	Adjusted (SD).	
CV MORTALITY								
PULITI [24]	1406	Undefined	19	624	4.76 [1.79–12.68]	–	1.56 (0.49) (not adjusted)	–
LESSMANN	5718	ICD I00–99	403	4318	3.27 [2.37–4.50]	2.05 [1.51–2.79]	0.72 (0.15)	Age, smoking, BMI, CV history, diabetes and hypertension
RASMUSSEN [25]	1945	ICD I00–99	19	910	3.19 [1.15–8.81]	2.26 [0.97–5.29]	0.82 (0.43)	Sex, age, smoker status, hypertension, hyper-cholesterolemia, diabetes
SHEMESH [26]	8782	ICD I00–78	150	5209	2.39 [1.70–3.35]	1.26 [0.95–1.67]	0.23 (0.14)	Sex, age, smoking history (no. of PY), diabetes
ALL-CAUSE MORTALITY								
JACOBS [21]	958	–	54	614	8.53 [2.10–34.67]	5.98 [2.49–14.35]	1.79 (0.44)	Sex, age, smoker status, hypertension diabetes, hyper-cholesterolemia
LESSMANN [22]	5718	–	1480	4318	1.97 [1.74–2.22]	1.43 [1.29–1.58]	0.36 (0.05)	Age, smoking status, BMI, CV history, diabetes and hypertension
RASMUSSEN [25]	1945	–	48	910	2.28 [1.41–3.68]	1.81 [1.17–2.81]	0.59 (0.22)	Sex, age, smoking status, hypertension, diabetes, hyper-cholesterolemia

Abbreviations: RR, relative risk; ICD, International Classification of Diseases; BMI, body mass index; PY, pack-years.

**Table 3 cancers-13-01553-t003:** Cardiovascular (CV) or all-cause mortality according to a high coronary artery calcification (CAC) score.

Study	*N*	Agaston Score	High CAC		RR (95% CI)		Log RR	Adjusted to
			Events	Total	Unadjusted	Adjusted	(σ log RR ajusté)	
CV MORTALITY								
LESSMANN [22]	5718	> 1000	–	–	–	2.76 [1.56–4.88]	1.02 (0.29)	Age, smoking status, BMI, CV history, diabetes and hypertension
RASMUSSEN [25]	1945	>400	5	132	7.84 [2.30–26.73]	3.8 [1.0–15]	1.34 (0.69)	Sex, age, smoking status, diabetes, hypertension, hypercholesterolemia
SHEMESH [26]	8782	Category 4–12	84	1640	4.26 [2.96–6.12]	2.1 [1.4–4.1]	0.74 (0.33)	Sex, age, smoking history (no. de PY), diabetes
ALL-CAUSE MORTALITY								
JACOBS [21]	958	>1000	24	137	16.99 [4.08–70.71]	10.93 [2.36–50.60]	2.39 (0.77)	Sex, age, smoker status, hypertension diabetes, hyper-cholesterolemia
LESSMANN [22]	5718	>1000	–	–	–	2.20 [1.44–3.36]	0.79 (0.21)	Age, smoking status, BMI, CV history, diabetes and hypertension
RASMUSSEN [25]	1945	>400	10	132	3.27 [1.60–6.68]	2.1 [1.0–4.8]	0.74 (0.41)	Sex, age, smoking status, diabetes, hypertension, hypercholesterolemia
SVERZELLATI [23]	1159	>400	5	80	7.49 [2.57–21.83]	3.73 [1.05–13.32]	1.32 (0.64)	Sex, age, smoking status, smoking duration, BMI, hypertension, diabetes

**Table 4 cancers-13-01553-t004:** Frequency of coronary artery calcifications (CACs) according to sex, when available.

	Men		Women		
	*n*	%	*n*	%	*p*-Value
CAC > 0					
Sverzellati [23]	442/793	56%	93/366	25%	0.001
Jacobs [21]	NA	NA	NA	NA	
Lessmann [22]	2955/3553	83%	1363/2165	63%	<0.001
Rasmussen [25]	644/1075	60%	266/870	31%	<0.001
Shemesh [26]	2975/4294	69%	2238/4488	49.9%	<0.001
Puliti [24]	NA	NA	NA	NA	
Total	4061/6764	60%	3960/7889	50%	<0.001
CAC > 400 or > category 4					
Sverzellati [23]	72/793	9%	8/366	2%	<0.001
Jacobs [21]	NA	NA	NA	NA	
Lessmann [22]	NA	NA	NA	NA	
Rasmussen [25]	105/1075	10%	27/870	3%	<0.001
Shemesh [26]	1062/4294	25%	578/4488	12.9%	<0.001
Puliti [24]	NA	NA	NA	NA	
Total	1239/6166	20%	613/5724	11%	<0.001

**Table 5 cancers-13-01553-t005:** Cardiovascular (CV) events and mortality according to sex.

Study	*N*	Type	Men		Women		Unadjusted RR	LOG RR
			Events	Total	Events	Total	(95% CI)	(σ log RR)
Sverzellati [23]	1159	CV event ^a^	30	793	3	366	4.62 [1.42–15.03]	1.53 (0.59)
Jacobs [21]	958	CV event ^b^	123	671	4	137	6.28 [2.36–16.71]	1.84 (0.49)
Lessmann [22]	5718	CV mortality ^c^	318	3553	125	2165	1.55 [1.27–1.89]	0.44 (0.10)
Rasmussen, [25]	1945	CV mortality ^c^	17	1075	2	870	6.88 [1.59–29.69]	1.93 (0.73)
Shemesh [26]	8782	CV mortality ^d^	122	4294	71	4488	1.80 [1.34–2.40]	0.59 (0.14)

Abbreviation: RR, relative risk. ^a^ Acute coronary syndrome, unstable angina, coronary revascularization. ^b^ International; Classification of Diseases, Ninth Revision Codes 424, 428, 430–438, 440, 441, 443–444. ^c^ ICD I00–I99. ^d^ ICD I00–I78.

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
