# Peer review of "Impact on All-Cause and Cardiovascular Mortality Rates of Coronary Artery Calcifications Detected during Organized, Low-Dose, Computed-Tomography Screening for Lung Cancer: Systematic Literature Review and Meta-Analysis"

_cancers, 2021, doi:10.3390/cancers13071553_

Round 1

Reviewer 1 Report

  1. ABSTRACT: The objectives are not clearly stated in the abstract.  “This study was undertaken to evaluate the impact of screening-scan–detected coronary artery calcifications (CACs) on CVD and overall mortality.”  This statement is incorrect; this is not a study of the impact of knowing CAC on CVD and mortality. Rather, this is a meta-analysis study determining the association between CAC detected on low dose screening CT and either CV mortality or all-cause mortality.
  2. ABSTRACT AND THROUGHOUT MANUSCRIPT: Terms like overall mortality or global mortality can create confusion.  Recommend using the term “all-cause mortality” consistently throughout the manuscript.
  3. ABSTRACT: What is the definition of CV events?  Why is CV events being compared between men and women without any context with regard to CAC scores?  What are the CAC differences between men and women and do they account for the differences in CVD mortality, all-cause mortality, or CV events?
  4. Please carefully define CV/CVD mortality, all-cause mortality, CV events to clarify terms for the reader. Please carefully define all other terms (MI, CVA, CV events, etc) throughout.
  5. Recommend revising the introduction.  Some of the statements are confusing as written.  Please consider carefully the context and goals of the study in the introduction.
    1. Example:“In theory, CVDs seem clearly amenable to organized screening with identified risk factors, modifiable at the individual level, and for which decreased exposure leads to decreased risk [9]; moreover, existence of an asymptomatic or latent phase during which interventions can effectively change the natural evolution of the disease [10].”  This sentence is 4 lines and confusingly written. 
    2. “Indeed, no test—with an acceptable cost—has demonstrated sufficient ability to identify subjects at risk for a CV event.”  This blanket statement is not true.  There are multiple tests that help identify subjects at risk for CV events; BP, Lipid Panel, hsCRP, calcium scoring, etc.
  6. Please expand upon differences between low dose, nonECG gated CT and dedicated cardiac ECG-gated CT studies, including factors that drive changes in the CAC scores between them in the introduction.
  7. METHODS:Many identified articles were from the literature were ultimately excluded, potentially leading to publication bias.  Please clarify in the methods and in the Figure 1 flow chart in more detail exactly what drove selections and exclusions at each stage.  How did the authors go from 1913 to 119, and then exactly why were 91 excluded leaving 28?
  8. METHODS:Please clarify the structure of data abstraction and what drove decisions about characteristics for data abstraction.  Were definitions of terms comparable between articles? Similarly, what was the definition of “active smoking” and was it the same across articles?
  9. METHODS:What was done based on information derived from methodological quality assessment?  In other words, were some articles removed?  How many?
  10. METHODS:  Please clarify inclusion/exclusion criteria for selected articles.
  11. METHODS:What was done based on information derived from methodological quality assessment?  In other words, were some articles removed?  How many?
  12. Limitations:  Please expand upon limitations.  A major limitation in this study is that simply knowing CAC may not readily translate to improved survival outcomes.  How would this information reclassify aging smokers, who are already high risk for CAD and would require optimal medical therapy. Is there an opportunity for reclassification of risk?  Is there an opportunity to incorporate information learned into a clinical trial with an intervention based on the CAC data? 
  13. CONCLUSION:Important to specify patient population of active and former smokers participating in lung cancer screening as the results are not broadly applicable to the general population.

Reviewer 2 Report

This is a review and meta-analysis of the mortality rate of cardiovascular disease. Low dose lung cancer screening has been beneficial to decrease lung cancer mortality. The authors collected data from published studies and showed that the presence of calcifications is associated with increased risk of mortality. My main comment is about the differences between this study and the  Evaluation of cardiovascular risk in a lung cancer screening cohort by Ruparel et al?

In line 132 the authors mention the Rit is not very clear which was the test. 

In line 142: What about the other model? How much was the RR? 

Why the specific models were chosen by the authors? 

The paper is written in a clear way, however there are a few typos. 

Round 2

Reviewer 1 Report

Thank you for the careful revisions.  I have no further comments.